# Linear Attention-based Multiple Instance Learning for Computational Pathology

**Charlotte Richter**[1*]                  Hedwig-Charlotte.Richter@stud.uni-regensburg.de

**Daniel Reisenbüchler**[1*]                       Daniel.Reisenbuechler@ur.de

**Nadine S. Schaadt**[2]                      Schaadt.Nadine@mh-hannover.de

**Friedrich Feuerhake**[2]                   Feuerhake.Friedrich@mh-hannover.de

**Dorit Merhof**[1,3]                                Dorit.Merhof@ur.de

[1]*Faculty of Informatics and Data Science, University of Regensburg, Germany*

[2]*Institute of Pathology, Hannover Medical School, Hannover, Germany*

[3]*Fraunhofer Institute for Digital Medicine MEVIS, Bremen, Germany*

## Abstract

Deep learning–based analysis of gigapixel whole slide images (WSIs) in computational pathology (CPath) typically relies on patch-level feature extraction and instance aggregation, with attention-based contextualization at the core of state-of-the-art methods. However, scalability is a major challenge due to the vast number of patches. Therefore, we introduce linear attention based multiple-instance learning (Lin-MIL), which transposes and interchanges the calculations of queries, keys, and values in the attention mechanism. By leveraging linear attention, Lin-MIL reduces computational complexity from $\mathcal{O}(n^2d)$ to $\mathcal{O}(nd^2)$, compared to vanilla self-attention. Despite this efficiency gain, Lin-MIL outperforms 12 baseline methods across biomarker, mutation, and tumor classification benchmarks, while also demonstrating robust out-of-domain performance. Moreover, its qualitative attention maps highlight diagnostically relevant regions. In summary, Lin-MIL provides increased performance as well as enhanced scalability and interpretability for a range of computational pathology tasks. Code available at `https://github.com/charlotterchtr/Lin-MIL`.

**Keywords:** Computational Pathology, Multiple Instance Learning, Linear Attention, Whole Slide Image Analysis

## 1 Introduction

Deep learning–based whole slide image analysis faces unique challenges due to the gigapixel scale of the data. To overcome the computational burden, the standard approach serializes WSIs into sequences of patch-level feature vectors using foundation models, followed by aggregation via multiple instance learning (MIL). Recent advances employ state-space models (Fillioux et al., 2023; Fang et al., 2024) or various self-attention mechanisms (Shao et al., 2021; Reisenbüchler et al., 2022; Tang et al., 2024; Wagner et al., 2023; Li et al., 2023; Xu et al., 2024) to contextualize these sequences. However, vanilla self-attention incurs quadratic complexity with respect to the number of sequence elements, limiting the number of patches that can be processed. This limitation is especially pronounced when

---

∗. These authors contributed equally to this work.

incorporating multiple intra-stained slides (e.g., multiple H&E slides), inter-stained slides (e.g., H&E and IHC stains) (Jaume et al., 2024; Reisenbüchler et al., 2024), or additional omics data (Vaidya et al., 2025) into MIL frameworks. Moreover, while Vision Transformers are typically applied to images up to 1024 pixels, WSIs contain orders of magnitude more patches, where $n \gg 1024$ and the latent dimension $d \leq 1024$. In such cases, the self-attention matrix is prone to a low-rank bottleneck (Li et al., 2024). Approaches like TransMIL (Shao et al., 2021) mitigate this by approximating self-attention via the Nyström method, though at the cost of performance. On the other hand, dilated attention based MIL methods (Xu et al., 2024) restrict calculations to local regions, impairing long-range dependency modeling. In this study, we introduce Lin-MIL, which leverages linear attention modules (Zheng, 2025) by interchanging the order of query and value computations and performing a transposed matrix multiplication. This reformulation reduces complexity from $\mathcal{O}(n^2 d)$ to $\mathcal{O}(n d^2)$ while capturing the most informative relationships through a $d \times d$ matrix that exploits the low-rank structure of the original attention matrix. Despite its linear complexity in the number of sequence elements, Lin-MIL outperforms 12 baseline models across eight computational pathology datasets, spanning biomarker, mutation, and metastasis prediction tasks. Moreover, Lin-MIL provides particularly notable gains in out-of-domain evaluations. Also, qualitative attention heatmaps further demonstrate that Lin-MIL reliably focuses on diagnostically relevant regions. Our main contributions are: (1) Lin-MIL, a novel MIL framework integrating linear attention modules, and (2) a comprehensive evaluation across multiple tasks, datasets, and methods, using foundation model-derived features, thus providing state-of-the-art benchmark results for WSI analysis.

## 2 Method

The overall design of our Lin-MIL pipeline is illustrated in Fig. 1. Our algorithm first transforms the WSI into a set of features, and then our Lin-MIL architecture aggregates these features to a slide-level prediction. In the following subsections, a detailed description of the process is provided.

**(A) Feature Embedding Stage.** We follow established standard preprocessing steps (Fig. 1A) and tesselate the WSI at $20\times$ magnification scale into $n$ smaller patches of size $\mathbb{R}^{512 \times 512 \times 3}$. These patches are subsequently passed to a pathology foundation model (FM) to extract patch-wise features. Thus, each WSI is given as a sequence $\{x_i\}_{i=1}^n \in \mathbb{R}^{n \times D}$, where $D$ represents the feature latent dimension.

**(B) Lin-MIL based aggregation.** In the aggregation stage (Fig. 1B), the sequence of patch-embeddings, $\{x_i\}_{i=1}^n$ with dimensionality $D$, is first projected to a lower-dimensional space $d$ via a fully connected (FC) layer. We append a classification token (CLS) to the sequence for information pooling, CLS $\in \mathbb{R}^{1 \times D}$. For the sake of brevity, the sequence length is denoted hereafter by $n$. Next, the sequence is processed through $l$ sequential linear attention blocks (Fig. 1C), comprising a linear attention module, accompanied by skip connections, normalization, and a final multi-layer perceptron.

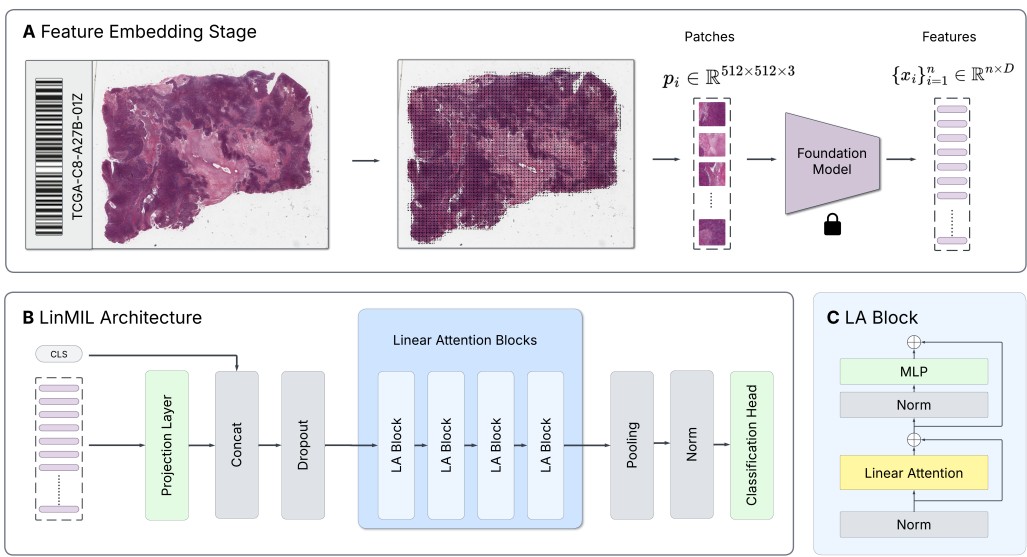

Figure 1: **Lin-MIL pipeline for WSI analysis. (A)** In the feature embedding stage, we tesselate the WSI after background removal and extract patch-level features using a pathology FM. **(B)** The Lin-MIL architecture shrinks the latent dimension by a projection layer, and aggregates the sequence by linear attention blocks followed by pooling and a classification head. **(C)** Each linear attention block calculates linear attention followed by normalization and a multilayer perceptron.

**(C) Linear Attention Module.** In the following, we derive linear attention from vanilla softmax attention (Fig. 2B), which is defined as:

$$V' = \mathrm{SoftA}_l(Q, K, V) = \mathrm{softmax}\left(\frac{QK^T}{\sqrt{d_k}}\right) V, \tag{1}$$

with queries $Q \in \mathbb{R}^{n \times d_k}$, keys $K \in \mathbb{R}^{n \times d_k}$, and values $V \in \mathbb{R}^{n \times d_v}$. Queries, keys, and values are derived from the input sequence $x$ as follows:

$$Q = W_Q \cdot x, \quad K = W_K \cdot x, \quad V = W_V \cdot x.$$

Thus, the softmax operation acts as a similarity function $\mathrm{Sim}(\cdot)$, returning the exponential of the dot product between queries and keys. Hence, self-attention for the $i$-th patch can also be expressed as

$$V_i' = \sum_{j=1}^{N} \frac{\mathrm{Softmax}(Q_i, K_j^T)}{\sum_{k=1}^{N} \mathrm{Softmax}(Q_i, K_k^T)} \quad V_j = \sum_{j=1}^{N} \frac{\mathrm{Sim}(Q_i, K_j^T)}{\sum_{k=1}^{N} \mathrm{Sim}(Q_i, K_k^T)} V_j, \tag{2}$$

where scaling factors are omitted for simplicity. To address the quadratic complexity of the above softmax attention, we follow (Katharopoulos et al., 2020) and replace the softmax

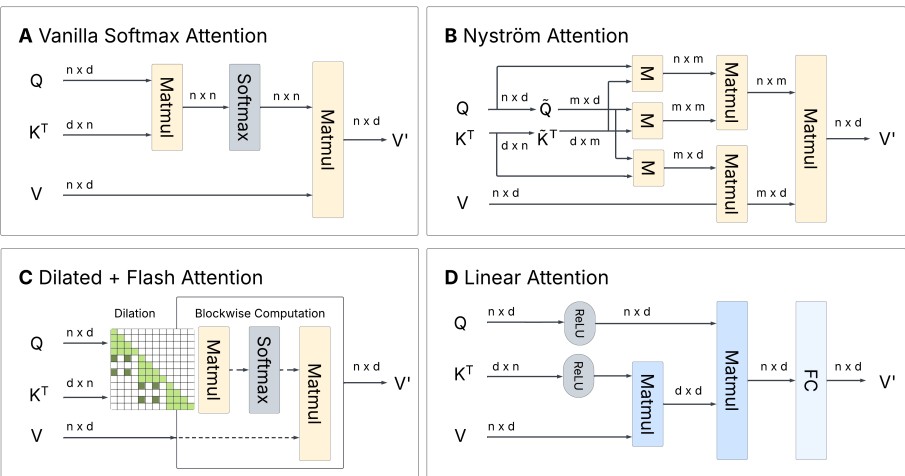

Figure 2: **Comparison of attention mechanisms. (A) Vanilla softmax attention** calculates attention scores by multiplying queries $Q$ and keys $K$ in $\mathcal{O}(n^2d)$, followed by softmax weighting and multiplication with values $V$. **(B) Nyström Attention** approximates self-attention by incorporating rank reduction to achieve a complexity of $\mathcal{O}(nm)$, with landmarks $m \ll n$. **(C) Dilated Attention** reduces the number of operations by varying dilation ratios to $\mathcal{O}(n)$. **(D) Linear Attention** uses a decomposable kernel function $\phi(\cdot) = ReLU(\cdot)$, to first calculate $\phi(K^T) \times V$ and then obtain the attention-weighted values $V'$ in $\mathcal{O}(nd^2)$.

function with a decomposable kernel function $\text{Sim}(\cdot) = \phi(\cdot)$,

$$\text{Sim}(Q_i, K_j^T) = \phi(Q_i)\phi(K_j^T).$$

By using the associative property of matrix multiplication, the self-attention term in Equation (2) can be re-written as:

$$V_i' = \frac{\sum_{j=1}^{N} \phi(Q_i)\phi(K_j^T)}{\sum_{k=1}^{N} \phi(Q_i)\phi(K_k^T)} \quad V_j = \frac{\phi(Q_i)\sum_{j=1}^{N} \phi(K_j^T)V_j}{\sum_{k=1}^{N} \phi(Q_i)\phi(K_k^T)}. \tag{3}$$

By separating the queries and keys, we first multiply the values and keys due to matrix associativity, and then multiply by $Q$ (Fig. 2D). This results in a reduction of the complexity from $O(n^2d)$ to $O(nd^2)$. Following Zheng (2025), we use the ReLU activation function as kernel $\phi(\cdot)$, which ensures non-negative values in the attention map. As in other attention variants, linear attention can be computed in parallel across multiple heads which are concatenated and linearly projected.

## 3 Experiments

We assess the performance of Lin-MIL on multiple CPath tasks. In the following, we present datasets, baselines, evaluation schemes, and implementation details.

### 3.1 Datasets and CPath Tasks

We predict microsatellite instability (MSI) in colorectal cancer using TCGA-CRC (Network, 2012) (N=447; 65 positive, 382 negative) and CPTAC-CRC (Edwards et al., 2015) (N=221; 53 positive, 168 negative) as training data. We externally validate on the PAIP cohort (N=47, 12 positive, 35 negative). We assess lymph node metastasis detection in breast cancer using the CAMELYON16 dataset (Bejnordi et al., 2016). Genetic alteration prediction for TP53 is performed for 4 different organs, in particular TCGA-BRCA (N=1114, 737 positive, 377 negative), TCGA-NSCLC (N=1026, 336 positive, 690 negative), TCGA-UCEC (N=549, 342 positive, 207 negative) and TCGA-STAD (N=413, 209 positive, 204 negative).

### 3.2 Evaluation and Comparable Methods

We conduct patient-stratified 5-fold cross-validation (CV) for each task and report results using the area under the receiver operating characteristic curve (AUROC), balanced accuracy (Bal. Acc), and weighted F1-Score. We benchmark Lin-MIL against MIL methods, including AB-MIL (Ilse et al., 2018) based on instance-wise attention, Transformer-MIL (Wagner et al., 2023) using softmax self-attention, CLAM-SB (Lu et al., 2021), which incorporates clustering-constrained attention, DSMIL (Li et al., 2021), which uses a dual-stream attention approach, LA-MIL (Reisenbüchler et al., 2022), which employs local graph-based attention, GTP (Zheng et al., 2022), a graph transformer, RRT-MIL (Tang et al., 2024) focusing on feature re-embedding, SC-MIL (Yang et al., 2024) using supervised contrastive learning, Long-MIL (Li et al., 2023), which integrates a linear bias into attention, S4MIL (Fillioux et al., 2023) and MamMIL (Fang et al., 2024), where both are structured state-space models, and TransMIL (Shao et al., 2021), which utilizes Nyström-based attention approximation. We use the same data preprocessing steps for all methods (Fig. 1A).

### 3.3 Implementation

Patch extraction was performed using the CLAM library (Lu et al., 2021) and subsequent feature extraction through the UNI FM (Chen et al., 2024). We addressed class imbalances during training by a weighted cross-entropy loss. We employed the ADAM optimizer with batch size 1, a learning rate of $1e-5$ and a weight decay of $1e-2$, for a maximum of 100 epochs with early stopping. We used a linear learning rate scheduling with a factor of $1e-1$ if performance plateaus occur. All experiments were executed on a single NVIDIA RTX 4500 with 25 GB GPU memory.

## 4 Results

### 4.1 Performance Analysis

Table 1 summarizes the 5-fold patient-stratified cross-validation performance of Lin-MIL and baseline methods on CAMELYON16. Additionally, we report MSI prediction results from models trained and validated on TCGA-CRC and CPTAC-COAD, and tested on the external PAIP cohort. Lin-MIL marginally outperforms all comparators on CAMELYON16 and shows significant generalization improvements on PAIP, with gains of +5% in balanced accuracy and +6% in weighted F1. Figures 3A-D present bar charts for TP53 mutation

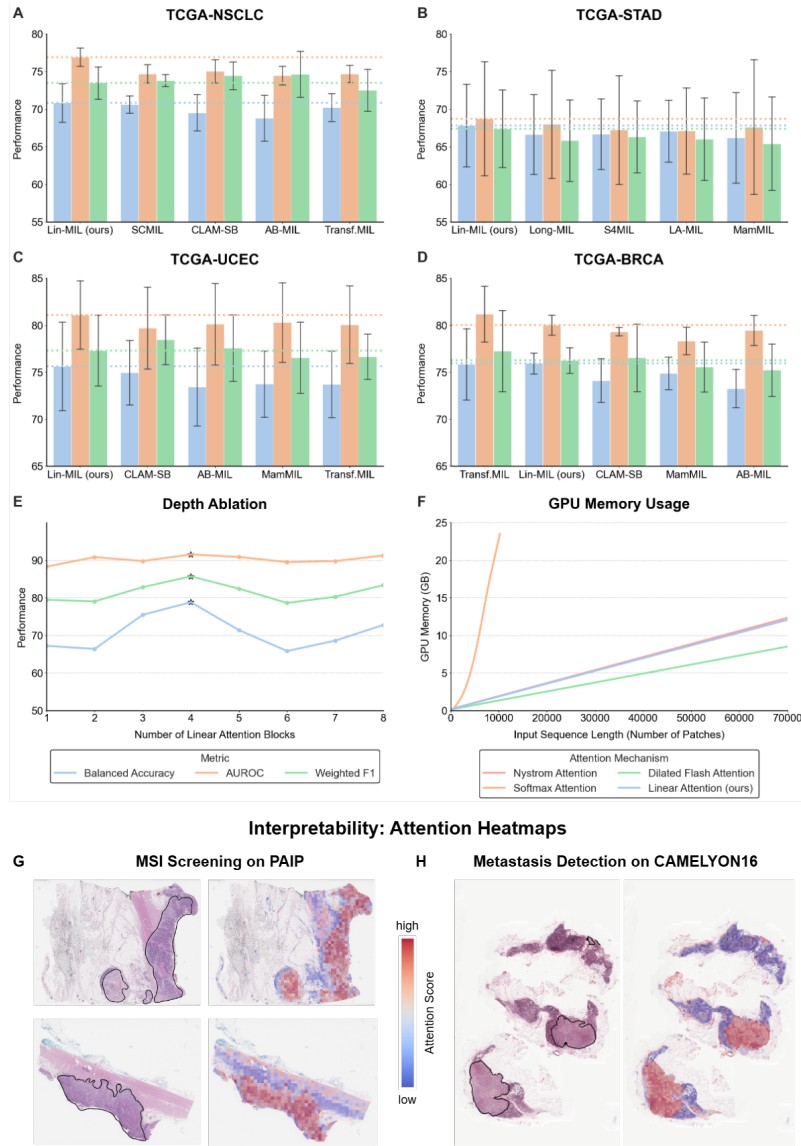

Figure 3: **Results. (A-D) Genetic mutation prediction for TP53:** Bars show mean and standard deviation for the top 5 performing models out of all listed in Table 1, horizontal lines visualize the mean performance of our Lin-MIL model. **(E) Depth ablation** on the number of linear attention blocks in our Lin-MIL model on MSI prediction trained on TCGA/CPTAC, and tested on PAIP. **(F) GPU memory usage** of linear attention compared to softmax, nyström and dilated attention. **(G-H) Lin-MIL attention heatmaps** for PAIP and CAMELYON16 datasets, annotations (left) and attention heatmaps (right) for two slides of PAIP and CAMELYON16 are displayed, with patches colored according to their normalized scores $V'$, see Equation 3.

Table 1: **Performance analysis.** We report AUROC, balanced accuracy (Bal. Acc) and weighted F1 (W. F1) metrics using mean and standard deviation over patient-stratified CV runs. Results for MSI Screening reported for external test cohort. Best in **bold** and second best is underlined.

| Task | MSI Screening | | | Metastasis Prediction | | |
|---|---|---|---|---|---|---|
| Model / Metric | AUROC | Bal. Acc. | W. F1 | AUROC | Bal. Acc. | W. F1 |
| AB-MIL | 90.57±1.6 | 61.67±5.4 | 75.23±4.9 | 97.93±1.9 | 91.96±10.9 | 92.84±9.9 |
| CLAM-SB | 90.86±1.4 | 64.71±6.7 | 77.52±5.7 | 98.44±1.9 | 96.51±2.0 | 97.02±1.7 |
| DSMIL | 89.48±0.5 | 73.29±10 | 74.69±6.1 | 87.45±16.1 | 86.13±15 | 87.20±14 |
| GTP | 82.71±5.2 | 71.60±3.8 | 73.41±11 | 85.62±9.4 | 82.64±6.4 | 84.10±5.7 |
| LA-MIL | 86.57±3.8 | 58.88±1.2 | 72.86±0.9 | 82.78±9.9 | 80.51±9.6 | 80.99±9.6 |
| Long-MIL | 87.52±4.2 | 61.93±7.6 | 74.95±7.1 | 97.53±2.7 | 95.73±3.2 | 96.26±2.7 |
| RRT-MIL | 89.43±2.9 | 63.05±9.1 | 75.77±7.9 | 98.32±1.0 | 95.68±3.1 | 95.57±3.3 |
| SC-MIL | 90.76±1.6 | 68.02±9.5 | 79.28±7.0 | 98.92±1.4 | 97.41±1.8 | 97.77±1.6 |
| TransformerMIL | 89.76±2.5 | 67.76±11 | 79.27±8.5 | 99.35±0.8 | 97.41±2.2 | 97.77±2.0 |
| TransMIL | 90.52±4.9 | 60.83±3.7 | 74.61±3.5 | 93.60±3.1 | 88.62±3.1 | 89.81±2.3 |
| S4MIL | 87.57±2.9 | 58.60±3.0 | 72.45±2.8 | 84.94±8.2 | 81.16±8.6 | 81.96±8.3 |
| MamMIL | 89.52±2.9 | 63.33±9.9 | 76.22±7.9 | 98.64±1.7 | 96.02±2.5 | 96.29±2.3 |
| **Lin-MIL** (ours) | **91.52±1.8** | **78.81±6.2** | **85.69±3.7** | **99.49±0.7** | **98.15±2.6** | **98.15±2.6** |

prediction across TCGA-NSCLC, TCGA-STAD, TCGA-UCEC, and TCGA-BRCA, where Lin-MIL ranks first in three out of four datasets (TCGA-STAD, TCGA-NSCLC, TCGA-BRCA) and second in TCGA-UCEC. Overall, Lin-MIL proves to be the most robust model across tasks with in average +0.4% balanced accuracy and +0.6% AUROC compared to the second best model, respectively. Finally, Figures 3G and 3H compare Lin-MILs attention heatmaps with ground-truth annotations for CAMELYON16 and PAIP, demonstrating its ability to focus on clinically relevant regions.

## 4.2 Ablation Study

We varied the number of linear attention blocks from 1 to 8 and identified 4 blocks as the optimal configuration for MSI prediction on TCGA-CRC and CPTAC-COAD (Fig.3E). Figure 3F illustrates GPU memory usage for four attention mechanisms: Linear attention (Lin-MIL), vanilla softmax attention (TransformerMIL), Neyström attention (TransMIL), and dilated attention (GigaPath). Figure 4A-B further compares these mechanisms (embedded within the same MIL architecture Fig. 1B), across four TP53 prediction tasks in TCGA cohorts, metastasis detection on CAMELYON16, and out-of-distribution MSI screening on PAIP. Although Neyström approximation exhibits the highest GPU memory usage among complexity-reducing approaches, it delivers only the second-lowest performance. In contrast, dilated attention uses the least memory but yields the poorest results. Lin-MIL, with slightly lower memory usage than Neyström attention, consistently excels across all datasets compared to memory efficient methods.

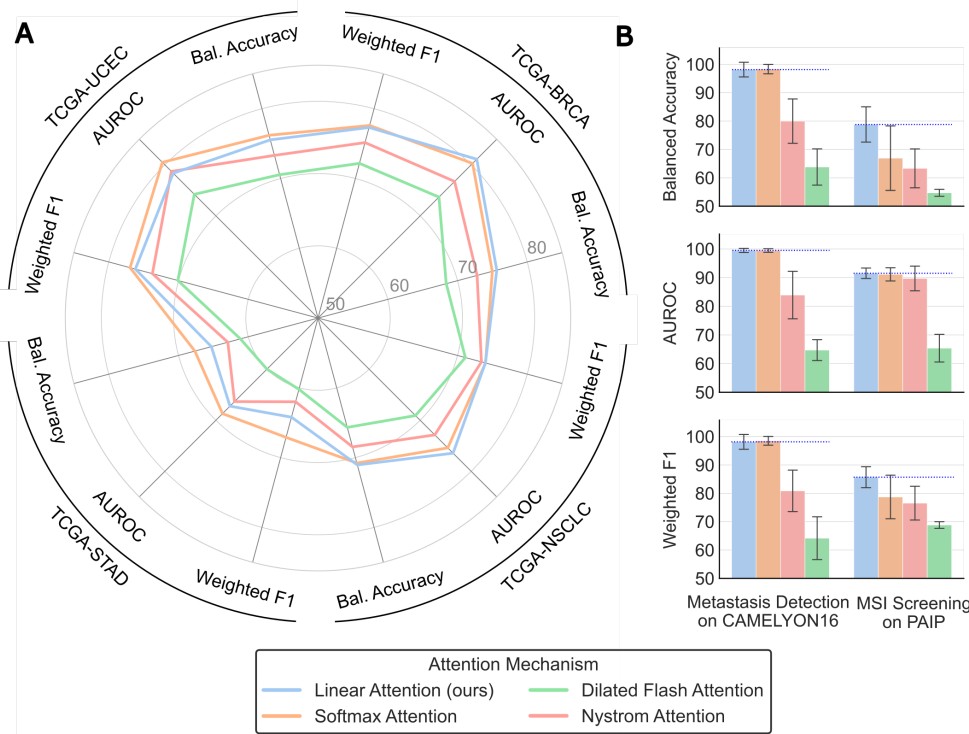

Figure 4: **Ablation on attention mechanisms.** We replaced linear attention with softmax-, Neyström- and dilated attention and report the mean performance over 5-fold CVs. **(A) TP53 mutation prediction results**, experiments setup as in Fig 3A-D. **(B) Metastasis and MSI prediction results**, experimental configurations as reported in Table 1.

## 5 Conclusion and Future Perspective

We presented Lin-MIL, a linear attention-based architecture designed to address computational challenges associated with large input sequences and limited memory scenarios in WSI analysis. By reducing complexity with respect to the number of patches to a linear scale, linear attention enables the processing of a large number of patches while providing interpretability and enhanced performance. In future work, we will study the behavior of linear attention in multi-modal settings for intra- and inter-modal feature fusion.

## Acknowledgments and Disclosure of Funding

This work was supported by the German Research Foundation (Deutsche Forschungsgemeinschaft, DFG) under project number 445703531. The authors gratefully acknowledge the computational and data resources provided by the Leibniz Supercomputing Centre (www.lrz.de).

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
