# OpenReview forum: "Linear Attention-based Multiple Instance Learning for Computational Pathology"
_MICCAI.org/2025/Workshop/COMPAYL — COMPAYL 2025_

### Official Review · Reviewer_7T7c · 2025-07-14
**Review of "Linear Attention-based Multiple Instance Learning for Computational Pathology"**

**Rating:** 3
**Confidence:** 3

**Review:**

The paper uses linear attention [1] instead of the usual softmax (which has quadratic complexity) in the Multiple Instance Learning(MIL) framework. The proposed method should decrease the computational burden without compromising performance.

The authors test the proposed solution on various tasks and cohorts.

The authors refer to [2] when explaining their linear attention mechanism; however, from the text, it seems that they only use ReLU as in [2] and the rest is the same as in [1]. This is difficult to judge because the authors do not make their code public.  Implementation of [2] also has LWA and LGA blocks.

The main paper[2] the manuscript is based on is a preprint that has not been peer reviewed yet.

In the introduction, variable n refers to the number of patches. In the method section, n is a different variable.
The Figure 3 legend is very confusing. No legend for bar plots, colors stand for different variables on the same figure.
The results are reported on 5-fold cross-validation. It seems like there was no separate test set used in some tasks?
On some tasks, methods with linear attention outperform the softmax one. It is unclear what the source of the performance gain is.


[1] Angelos Katharopoulos, Apoorv Vyas, Nikolaos Pappas, and Fran¸cois Fleuret. Transformers are rnns: fast autoregressive transformers with linear attention. In Proceedings of the 37th International Conference on Machine Learning, number 478 in ICML’20, pages 5156– 5165. JMLR.org, 2020.

[2] Chuanyang Zheng. The linear attention resurrection in vision transformer, 2025

---

### Official Review · Reviewer_u26i · 2025-07-15
**Well-executed work**

**Rating:** 5
**Confidence:** 5

**Review:**

This is a very well-written paper that addresses one of the major issues in applying self-attention-based multiple instance llearning (MIL) methods to WSIs, specifically, the quadratic complexity. Although the linear attention module itself is based on existing work and is not novel, the execution of the approach is excellent, clearly demonstrated by results where the Lin-MIL method consistently outperforms numerous baseline methods across diverse computational pathology tasks. Additionally, the paper includes ablation studies that effectively illustrate the importance of some of the design choices. Overall, the manuscript is easy to follow, logically structured, and provides solid evidence supporting the proposed method. One weak point is that the code is not publicly available and I encourage the authors to do so.